# *N^6^*-Methyladenine Progressively Accumulates in Mitochondrial DNA during Aging

**DOI:** 10.3390/ijms241914858

**Published:** 2023-10-03

**Authors:** Ádám Sturm, Himani Sharma, Ferenc Bodnár, Maryam Aslam, Tibor Kovács, Ákos Németh, Bernadette Hotzi, Viktor Billes, Tímea Sigmond, Kitti Tátrai, Balázs Egyed, Blanka Téglás-Huszár, Gitta Schlosser, Nikolaos Charmpilas, Christina Ploumi, András Perczel, Nektarios Tavernarakis, Tibor Vellai

**Affiliations:** 1Department of Genetics, Eötvös Loránd University, Pázmány Péter sétány 1/C, 1117 Budapest, Hungary; himanisharma1306@gmail.com (H.S.); egyed.balazs@ttk.elte.hu (B.E.);; 2Genetics Research Group, Eötvös Loránd Research Network-Eötvös Loránd University, 1117 Budapest, Hungary; 3Momentum Ion Mobility Mass Spectrometry Research Group, Hungarian Academy of Sciences-Eötvös Loránd University, 1117 Budapest, Hungary; 4Institute of Molecular Biology and Biotechnology, Foundation of Research and Technology-Hellas, P.O. Box 1385 Heraklion, Greece; 5Department of Organic Chemistry, Eötvös Loránd University, 1117 Budapest, Hungary; 6Vellab Biotech Ltd., 6722 Szeged, Hungary

**Keywords:** age determination, aging, *C. elegans*, dog, Drosophila, mitochondrial genome, N6-adenine methylation, N6-methyladenine levels

## Abstract

*N^6^*-methyladenine (6mA) in the DNA is a conserved epigenetic mark with various cellular, physiological and developmental functions. Although the presence of 6mA was discovered a few years ago in the nuclear genome of distantly related animal taxa and just recently in mammalian mitochondrial DNA (mtDNA), accumulating evidence at present seriously questions the presence of *N^6^*-adenine methylation in these genetic systems, attributing it to methodological errors. In this paper, we present a reliable, PCR-based method to determine accurately the relative 6mA levels in the mtDNA of *Caenorhabditis elegans*, *Drosophila melanogaster* and dogs, and show that these levels gradually increase with age. Furthermore, *daf-2(−)*-mutant worms, which are defective for insulin/IGF-1 (insulin-like growth factor) signaling and live twice as long as the wild type, display a half rate at which 6mA progressively accumulates in the mtDNA as compared to normal values. Together, these results suggest a fundamental role for mtDNA *N^6^*-adenine methylation in aging and reveal an efficient diagnostic technique to determine age using DNA.

## 1. Introduction

Understanding how the aging process is regulated remains a fundamental problem in biology, with significant medical, social and economic implications [1,2]. It has been known for decades that epigenetic changes, such as de novo DNA methylation and demethylation mediated by specific DNA methyltransferases and demethylases, are associated with aging [3,4,5,6]. During DNA methylation, a methyl group (-CH_3_) is added to a specific nucleobase, cytosine or adenine, primarily converting the former to *5*-methylcytosine (5mC) and the latter to *N^6^*-methyladenine (6mA). These modified nucleobases often alter the activity of the affected genetic locus (in general, 5mC represses, while 6mA promotes, gene expression), and the new DNA methylation pattern can be inherited by daughter cells and offspring for certain generations [7].

Efforts to understand the relationship between 5mC and aging have reached such an advanced stage in the last 10 years that several research groups have managed to set up a so-called epigenetic (also called Horvath) clock to estimate an individual’s age based on CpG methylation distributions in the genome [8]. Although relatively good results (with a deviation of ±7–10 years) have been obtained by predicting biological age from 5-cytosine methylation, the method still relies on a genome-wide methylation profile for which the 5mC pattern of the whole genome or at least significant parts of the genome has to be revealed, and this makes the method cumbersome, costly and slow.

DNA methylation at the *N^6^* position of adenine is the other major type of epigenetic modification, which has been widely recognized in bacteria and plants [9,10]. A few years ago, DNA 6mA modification was also identified in the genome of a diverse range of animal taxa ranging from worms to mammals [11,12,13,14,15]. Furthermore, the presence of 6mA was recently detected in mammalian mtDNA [16]. Accumulating evidence, however, seriously disputes the existence of *N^6^*-adenine methylation in animal and organellar genomes, suggesting that 6mA is not present in these genetic systems, and what was found previously is merely an artifact of bacterial DNA or RNA contamination or limitations in methods used for 6mA detection, including single-molecule real-time sequencing (SMRTseq), liquid chromatography-tandem mass spectrometry (LC-MS/MS) and hybridization with a 6mA-specific antibody [17,18,19,20].

The mitochondrion, a membrane-bound, energy-converting organelle of eukaryotic cells, is known to be involved in the regulation of the aging process across a wide variety of animal species; *Caenorhabditis elegans* (nematode), *Drosophila melanogaster* (insect) and mouse (mammalian) strains with decreased mitochondrial activity exhibit a long-lived phenotype [21,22,23,24,25]. It was recently demonstrated that, in competition with chromatin instability, decline in the integrity and function of mitochondrial DNA, which is triggered by a programmable cellular process, can per se determine aging in individual yeast cells [26]. However, the molecular mechanisms underlying the disintegration of the mitochondrial genome during aging remain largely unknown.

Because there is a strong association between the epigenetic modifications of genomic DNA and biological age [27], epigenetic modifications in the mitochondrial genome may be similarly related to the age of the organism, but this has not yet been investigated and explored. In this study, we present a novel, reliable, PCR- (polymerase chain reaction) based (i.e., sequence-specific) 6mA detection method that is free of technological artifacts and show in several genetic models that relative 6mA levels at different mtDNA sites (these levels actually show that how many percent of the individual mitochondrial genomes present in a given tissue sample are methylated at a selected adenine nucleobase) are significantly related to the age of the organism. Thus, *N^6^*-adenine methylation is an inherent process in the organization of mitochondrial genomes too. These results suggest that the widely observed age-related decline in mitochondrial function [28] is strongly associated with changing 6mA levels and that biological age can be accurately determined from 6mA levels at certain mtDNA sites in a reliable, fast and cost-effective way. Furthermore, we reveal the enzymatic pathways of the mtDNA *N^6^*-adenine methylation and demethylation processes in *C. elegans* and *Drosophila*, showing the involvement of DNA *N^6^*-adenine methyltransferases and *N^6^*-methyladenine demethylases mediating 6mA metabolism in the nuclear genome.

## 2. Materials and Methods

### 2.1. In Vitro Comparison of Different Methods in Determining Relative N^6^-Methyladenine Levels

L4440 served as the original RNAi vector (equivalent to pPD129.36; Fire Lab 1997 Vector Kit) [29]. The 344 bp-long MCS (Multi Cloning Site) fragment from L4440 was amplified by using left and right primers that contain T7 terminator sequences at both ends, together with 5′ PciI- and 3′ NgoMIV-specific restriction sites. PCR fragment was ligated to L4440 to create a modified RNAi vector, T444T [30]. The *Tc3* genomic fragment was cloned into T444T by using the following restriction enzymes, as well as forward and reverse primers: (BglII + Acc65I) 5′-TTT TTT AGA TCT ATG CCT CGA GGA TCT GCC CT-3′ and 5′-TTT TTT GGT ACC GGG TAA GTC TTG TTC TGA GCA TAC ACG-3′. Competent *E. coli* DH5α bacteria were transformed with the T444T vector containing the *Tc3* transposable element. Plasmid DNA was isolated using a commercial kit QIAprep (Qiagen, Hilden, Germany), following the manufacturer’s instructions. Since the DH5α strain contains all the enzymes necessary for adenine methylation, all vector DNAs isolated from the bacterium were considered fully methylated. The unmethylated DNA sections were created by PCR amplification, in which the template DNA was the *C. elegans* genomic DNA in order to avoid contamination from the template DNA, which could affect the adenine methylation level of the product. The genomic DNA was isolated from *C. elegans* according to the standard protocols (Thermo Scientific GeneJET Genomic DNA Purification Kits #K0721 and #K0722, Waltham, MA, USA). The following primers were used to create the unmethylated PCR product DNA: 5′-ATT GCT CAA TGT GTC CCT GC-3′ and 5′-CGT TTG ATG ACA TTG AGG ATG GTC C-3′. PCR conditions were as follows: initial denaturation at 95 °C for 2 min, followed by 35 cycles of denaturation at 95 °C for 30 s, annealing at 57 °C for 30 s and extension at 72 °C for 1 min.

Comparing different methods, we used the following enzymatic treatments, primers and qPCR protocols. For control qPCR primers we used: 5′-TTC GTG CTG CCT CCA ACT CCT G-3′ and 5′-CGT TTG ATG ACA TTG AGG ATG GTC C-3′ with the qPCR conditions: 95 °C for 30 s, then 95 °C for 10 s and 56 °C for 30 s, repeated for 60 cycles. The following PCR primers were used for the DpnI- and MboI-based assays, 5′-ATT GCT CAA TGT GTC CCT GC-3′ and 5′-CGT TTG ATG ACA TTG AGG ATG GTC C-3′, with the qPCR conditions: 95 °C for 30 s, then 95 °C for 10 s and 56 °C for 30 s, repeated for 60 cycles. The samples were digested with MboI or DpnI at 37 °C for 20 min, and then the enzymes were inactivated at 80 °C for 20 min. After the heat inactivation, the PCR experiments were performed. The following PCR primers were used for the DpnI- and subsequent linker ligation-based assay experiments, 5′-ATG AAA CTC TCT TAC CGT TAG GTC AGA TCT ATC C-3′ and 5′-CGT TTG ATG ACA TTG AGG ATG GTC C-3′, with the qPCR conditions: 95 °C for 30 s, then 95 °C for 10 s and 56 °C for 30 s, repeated for 60 cycles. The samples were digested with DpnI at 37 °C for 20 min, and then the enzyme was inactivated at 80 °C for 20 min. A specific linker DNA (adaptor) was subsequently ligated to the DNA fragments at 4 °C for overnight. After heat inactivation at 65 °C, the PCR experiments were performed. Forward and reverse primers, and PCR conditions were as follows. For the control (template copy number determination): 5′-TTC GTG CTG CCT CCA ACT CCT G-3′ and 5′-CGT TTG ATG ACA TTG AGG ATG GTC C-3′. Primers used for linker-free, MboI or DpnI digestion variations: 5′-ATT GCT CAA TGT GTC CCT GC-3′ and 5′-CGT TTG ATG ACA TTG AGG ATG GTC C-3′. Primers used for linker ligation-based assay: 5′- ATG AAA CTC TCT TAC CGT TAG GTC AGA TCT ATC C -3′ and 5′- CGT TTG ATG ACA TTG AGG ATG GTC C -3′. A total of 10 pg-1 µg DNA template was used. PCR conditions were: 95 °C for 30 s, then 95 °C for 10 s and 56 °C for 30 s, repeated for 50 cycles.

### 2.2. Determination of the Relative N^6^-Methyladenine Levels at Different mtDNA Sites

Genomic DNA was isolated from *C. elegans*, *Drosophila* and dog (blood) at different adult stages, according to the standard protocols (Thermo Scientific GeneJET Genomic DNA Purification Kits #K0721 and #K0722). The samples were digested with DpnI at 37 °C for 20 min, and then the enzyme was inactivated at 80 °C for 20 min. A specific linker DNA (adaptor) was subsequently ligated to the genomic DNA fragments at 4 °C overnight. After heat inactivation at 65 °C for 20 min, the PCR experiments were performed. Forward and reverse primers, and PCR conditions were as follows. For *C. elegans*, the control: 5′-TCA GTA TAT TTG ATT ATC AAT TTT AGC CAT TAT AAT AG-3′ and 5′-ATC TTT TGC GCT TAA AAC AAA TG-3′; Ce_mito3: 5′-CTT ACC GTT AGG TCA GAT CTA TCT AT-3′ and 5′-GAC AAA AAT TAA AAG AGC AGG AGT-3′; and Ce_mito4: 5′-CCG TTA GGT CAG ATC TAT CAA A-3′ and 5′-CTA CTA AGC CTT CTC CTC TC-3′. A total of 10 pg-1 µg DNA template was used. The PCR conditions were: 95 °C for 30 s, then 95 °C for 10 s and 56 °C for 30 s, repeated for 30/50 cycles. Linker DNA: 5′-TAT TAT GAA ACT CTC TTA CCG TTA GGT CAG ATC TA-3′. The following strains were used: Bristol (N2) as the wild type, CB1370 *daf-2(e1370)III*, VC2552 *nmad-1(ok3133)III* and VC40319 *C18A3.1/damt-1(gk961032)II*. Prior to performing 6mA assays, the VC2552 and VC40319 strains were isogenized by outcrossing 3 times with the wild type. For *Drosophila*, the control: 5′-GAT ACC TTT AAA AAC GAA TAA CAT TTC-3′ and 5′-CGT TAG GTC AGA TCT ATC AAG-3′; Dm_mito2: 5′-TAC CAA AAT ACT CCG CCA GC-3′ and 5′-ACC GTT AGG TCA GAT CTA TCA TG-3′; and Dm_mito4: 5′-CTT ACC GTT AGG TCA GAT CTA TCT TT-3′ and 5′-AAT GGA GGT AAT CCT CCT AAT G-3′. The PCR conditions were: 92 °C for 30 s, then 92 °C for 10 s, 60 °C for 30 s and 72 °C for 20 s, repeated for 20/36 cycles. Linker DNA: 5′-ATC TAG ACT GGA TTG CCA TTC TCT CAA AGT ATT AT-3′. Strain *w1118* was used as the control. For dog, the control: 5′-ATA GCT GGT TAC CCA CAG AC-3′ and 5′-GTT GGG TTA ACA ATG GGG TG-3′; and Cl_mito2: 5′-CTT ACC GTT AGG TCA GAT CTA TCA TC-3′ and 5′-GCT GGC CTA GTA GAG CAA AG-3′. The PCR conditions were: 95 °C for 30 s, then 95 °C for 10 s and 54 °C for 45 s, repeated for 25/35 cycles. Linker DNA: 5′-TAT TAT GAA ACT CTC TTA CCG TTA GGT CAG ATC TA-3′.

### 2.3. qPCR Quantification of the Relative mtDNA 6mA Levels in Canine Frontal Cortex

Quantitative real-time PCR reactions were performed in a Roche LightCycler 96 Instrument (Roche Molecular Systems, Pleasanton, CA, USA) with FastStat Essential DNS Green Master kit (Roche, 06924204011). The quantitative measurements were repeated three times, and each qPCR experiment contained three technical repeats. A mitochondrial DNA fragment level was used as the internal control. Forward (F) and reverse (R) primers were as follows: Control F: 5′- ATA GCT GGT TAC CCA CAG AC-3′ and R: 5′- GTT GGG TTA ACA ATG GGG TG-3′, mitochondrial 6mA-specific primers: 5′- CCG TTA GGT CAG ATC TAT CCG a -3′ and 5′-TAA TGC CGG TAG GAG GTC AG-3′. The PCR conditions were: 95 °C for 30 s, then 95 °C for 10 s and 56 °C for 45 s, repeated for 30/50 cycles.

### 2.4. SMRT Sequencing and Data Analysis

Wild-type animals collected from 200–200 NGM (nematode growth medium) plates were harvested at the adult stages of days 1 and 5. Genomic DNA was isolated by GeneJET Genomic DNA Purification Kit (Thermo Scientific™ GeneJET Genomic DNA Purification Kit, #K0721) and then subjected to SMART sequencing. The command line tools of PACBIO SMRT Link (version 7.0.1) for methylation calling were used. First PacBio sequences (stored in unaligned BAM files) were mapped to the *C. elegans* reference genome (Caenorhabditis_elegans.WBcel235.dna.toplevel.fa) by pbalign (version 0.4.1). Then, ipdSummary (version 2.4) was used on the aligned BAM files to detect DNA base modifications from kinetic signatures. The methylation levels of sequences were calculated by in-house Python scripts using the gff files and the annotation files of Dfam release 3.1 (https://www.dfam.org/releases/Dfam_3.1/annotations/ce10/ (accessed on 2 October 2023), https://doi.org/10.1093/nar/gkv1272 (accessed on 2 October 2023)). To determine the relative 6mA content for mtDNA sequences, the amount of 6mA signals detected in the given section was divided by the length of the segment to obtain the relative amount of 6mA in that segment.

### 2.5. LC-MS/MS Analysis

mtDNA was isolated from wild-type (strain N2) *C. elegans* hermaphrodites at adult stages of 1 and 7 days (BioVision Mitochondrial DNA Isolation Kit, K280-50), and then digested by DNA Degradase Plus enzyme to nucleotides. The samples were incubated at 37 °C for 3 h, then transferred onto Pall Laboratory column (NanoSep 3 kDa, 29300-606) and centrifuged at 8000 rpm for 15 min. The concentrations of bases flowing through the column were measured. The measurements were performed on a high-resolution hybrid quadrupole-time of flight mass spectrometer (Waters Select Series IMS, Waters Corp., Wilmslow, UK) equipped with Z-spray electrospray ionization source. UHPLC separation was performed using a Waters Acquity I-Class UHPLC system. The linear gradient elution of eluent A (0.1% formic acid in water) and B (0.1% formic acid in acetonitrile) was used with the following gradient profile: 0 min: 2% B; 1 min: 2% B; 8 min: 60% B. The flow rate was set to 200 µL/min, and column temperature was set to 40 °C. The samples were dissolved in water. A total of 10 µL digested DNA was injected into the column. The Waters Acquity Premier Peptide BEH C18, 130 Å, 1.7 µm reversed phase column was used for the chromatographic separation. The MS/MS fragmentation of *m/z* 266.1 was performed in the transfer cell of the mass spectrometer using 17 V collision voltage. The optimized tune settings of the mass spectrometer are available in the Appendix A.

### 2.6. Generation of a Translational Fusion NMAD-1::GFP Reporter

A strain transgenic for a _p_nmad-1::NMAD-1::GFP translational fusion reporter was generated as follows. A 2.37 kb-long genomic fragment containing 5′ regulatory and coding regions of *nmad-1* was PCR-amplified using the following primers: 5′-AAA ACT GCA GGG CCC TTT CCT AGT TTT TGC-3′ and 5′-AAA CCC GGG ATT TCC CCA AAT CCA CAT ATC A-3′. The PCR fragment was cloned into the pPD95.75 vector at PstI and SmaI sites. Using a Biolistic PDS-1000/He particle delivery system (BioRad), transgenic animals were generated by microparticle bombardment (biolistic transformation) into the *unc-119(−)* (uncoordinated) mutant genetic background. The *unc-119(+)* gene was used as a co-transformation marker. Transgenic lines were established by picking up F1 animals with normal movement. A total of 10–15 μg linearized plasmid DNA was bombarded onto *unc-119(ed3)* mutant hermaphrodites at young adult stages.

### 2.7. TMRE Staining

TMRE (Tetramethylrhodamine, ethyl ester, perchlorate, catalog number T-669; Molecular Probes, Invitrogen, Waltham, MA, USA) was added on top of standard NGM plates seeded with OP50 bacterial food at a final concentration of 0.15 uM per plate. NMAD-1::GFP reporter animals synchronized at the L4 larval stage were placed on dye-containing NGM plates. Then, 48 h later, day 2 adult worms were anesthetized in a 20 mM tetramisole drop on microscopy slides, sealed with coverslips and analyzed with a Zeiss LSM 710 confocal microscope.

### 2.8. Isolation of the Mitochondria

For mitochondrial fractionation, we used a previously described protocol [31] and adapted it accordingly. A large amount (approximately 300–500 uL worm pellet volume) of synchronized day 1 adult animals of the wild type (N2) and NMAD-1::GFP translational reporter strains were washed off NGM plates using M9 buffer. After two washes with M9 (centrifugations for 1 min at 50× *g*), worm pellets were incubated in M9 supplemented with 10 mM Dithiothreitol (DTT) for 30 min at 4 °C with rotation. DTT was washed off the pellets three times with M9 buffer (centrifugations for 1 min at 200× *g* at 4 °C). The worm pellets were then homogenized by hand (approximately 100 strokes) with a 3 mL Potter-Elvehjem homogenizer with PTFE pestle and glass tube (Sigma-Aldrich) in isolation buffer (50 mM Tris-HCl pH 7.4, 210 mM mannitol, 70 mM sucrose, 0.1 mM EDTA, 2 mM PMSF and complete mini protease inhibitor cocktail (Roche), the last two being added just before use). Lysates were centrifuged at 300× g for 1 min at 4 °C and the supernatant fractions that included mitochondria were kept separately. The pellets were re-suspended in isolation buffer and homogenized as previously described. The homogenates were centrifuged at 300× g for 1 min at 4 °C. For each sample, the supernatants were combined with the ones from the previous step and centrifuged again at a low speed (300× *g*) and then at a high speed (13,000 rpm). The supernatants from the last centrifugation were stored as the cytoplasmic fractions, while the pellets were re-suspended in isolation buffer and stored as the mitochondrial fractions. For Western blot analysis, cytoplasmic and mitochondrial fractions were mixed with a Laemmli sample buffer with β-mercaptoethanol (final concentration 1×) and boiled for 5 min.

### 2.9. Western Blotting

Mitochondrial and cytoplasmic fractions, which were previously boiled in the Laemmli sample buffer, were analyzed by 10% SDS polyacrylamide gel electrophoresis (SDS-PAGE), were transferred on nitrocellulose membrane and blotted against various antibodies. We used anti-GFP (rabbit polyclonal, EnzyQuest, Vassilika Vouton, Greece), anti-MTCO1 (mouse monoclonal, abcam, Cambridge, UK) and anti-α-tubulin (mouse monoclonal, Develomental Studies Hybridoma Bank, Iowa City, IA, USA) antibodies.

### 2.10. Downregulation of Drosophila Tet and Mt2

*Drosophila* stocks were maintained in a standard cornmeal–sugar–agar medium at 18–25 °C, and experiments with flies were conducted at 25 °C or 29 °C as indicated. The strains were obtained from the Bloomington *Drosophila* Stock Center (BDSC) or kindly provided by other researchers. The following stocks were used: *Oregon-R*, wild type (a gift of Rita Sinka, University of Szeged, Szeged, Hungary); *w^1118^* (BDSC 5905); Tub-Gal4 (BDSC 5138); *Tet-RNAi* (BDSC 62280); *Mt2-RNAi^HMS01667^* (BDSC 38224); and *Mt2-RNAi^HMS02599^* (BDSC 42906). For RNA isolation, cDNA synthesis and semi-quantitative PCR analysis, the following was conducted: heads were dissected from 10–12 male adults (10 days old) in PBS, collected in TRI Reagent^®^ solution (Zymo Research, R2050-1-50, Sunnyvale, CA, USA) and homogenized. RNA isolation was performed according to the Direct-zol™ RNA MiniPrep kit (Zymo Research, R2050) protocol. External DNase treatment was conducted with DNAse I (Thermo Fisher Scientific, EN0523). Reverse transcription was performed by using the RevertAid First Strand cDNA Synthesis Kit (Thermo Fisher Scientific, K1621). For semi-quantitative PCR analysis, 100 ng of cDNA samples was used in a total reaction volume of 15 μL. The PCR mix was composed of 1.5 μL 10× DreamTaq Green Buffer (Thermo Scientific, EP0712), 0.15 μL DreamTaqTM Green DNA Polymerase (Thermo Scientific, EP0712), 10 mM of each dNTP and gene-specific oligonucleotide primers (5 μM). The following forward and reverse primers were used: for *Gapdh*, 5′-AAA AAG CTC CGG GAA AAG G-3′ and 5′-AAT TCC GAT CTT CGA CAT GGC-3′; for *Tet*, 5′-GAC GAA CAG TTC CAC GTC CT-3′ and 5′-CAG GGA AAT TTG TCC AGC AT-3′; and for *Mt2*, 5′- GAC AAA TAG TTG CCG CCT TG-3′ and 5′-CTG GGA TCA GTC CAC ACA GA-3′. The applied cycling parameters were initial denaturation at 95 °C for 2 min, denaturation at 95 °C for 30 s, an annealing temperature of 59 °C for 30 s, extension at 72 °C for 1 min (36 cycles) and a final extension at 72 °C for 5 min.

### 2.11. Quantifying N^6^-Methyladenine Levels in Canine mtDNA by qRT-PCR

Genomic DNA was isolated from blood samples. Quantitative real-time PCR reactions were performed in a Roche LightCycler 96 Instrument (Roche Molecular Systems) with the FastStat Essential DNS Green Master kit (Roche, 06924204011). Quantitative measurements were repeated three times, and each experiment involved three technical repeats. A mtDNA fragment level was used as the inner control. The forward (F) and reverse (R) primers were as follows: *control* F: 5′- ATA GCT GGT TAC CCA CAG AC-3′; *control* R: 5′- GTT GGG TTA ACA ATG GGG TG-3′; *Ca_mito2* F: 5′- CCG TTA GGT CAG ATC TAT CCG a -3′; and *Ca_mito2* R: 5′-TAA TGC CGG TAG GAG GTC AG-3′.

### 2.12. Statistical Analysis

Statistics for semi-qPCR and qPCR results were calculated using IBM SPSS Statistics v22. A Shapiro–Wilk test was used to assess whether there was a normal distribution of the examined samples. Every sample had a normal distribution; therefore, a one-way ANOVA with Tukey’s post hoc test or an independent *t*-test (with Levene’s test) was performed to compare the samples. Significance level marking: *p* < 0.05, *: *p* < 0.01, ***: *p* < 0.001 significance. NS marks no significance.

## 3. Results

### 3.1. Reliable PCR-Based Identification of Relative 6mA Levels at Different mtDNA Sites

Several techniques have been developed to identify DNA 6mA modification in viral, prokaryotic and eukaryotic genomes. These innovations include SMRTseq, LC-MS/MS and 6mA-specific antibody staining [11,12,13,16]. In recent years, however, the reliability of these methods has been seriously questioned [17,18,19,20]. Recent findings suggest that SMRTseq frequently misidentifies 5mC as 6mA [19]. Moreover, the contamination of bacterial DNA or endogenous RNA that also contain 6mA alterations can frequently lead to artifacts, which makes these current methods unreliable.

To overcome this problem, the methylation-sensitive or -dependent restriction endonucleases and a subsequent PCR amplification of the digested/undigested DNA fragments are used to quantify the level of methylated vs. unmethylated adenine nucleotides at a selected genomic site (*16*). DpnI, for example, is a methylation-dependent restriction enzyme that digests a GATC tetranucleotide target sequence when A is methylated (GA^Me^TC). A subsequent PCR reaction actually uses the undigested (intact), unmethylated genomic fragments as a template for DNA amplification (Figure 1A,A’). The problem arises from the fact that the methylation level of the selected adenine is usually very low; thus, even a significant difference in 6mA levels between two samples remains largely undetectable by PCR (if two tissue samples each contain 1000 somatic cells—2000 individual haploid genomes—and only two adenines are methylated at a selected genomic site in one of the samples and four adenines are methylated at the same site in the other sample, the difference is twofold, but the PCR reaction hardly distinguishes between 1998 and 1996 intact—undigested—genomic fragments; so, in this case, the difference is largely negligible). Alternatively, MboI is a methylation-sensitive restriction enzyme that cuts the GATC sequence when A is unmethylated (Figure 1A’’’). In this case, the methylated sequence (GA^Me^TC) remains intact and serves as a template for PCR amplification. In this case, the problem happens that not all of the GATC sequences at the selected genomic site become digested by the enzyme, and the remaining few intact, unmethylated (GATC) fragments compete with the intact, methylated ones (GA^Me^TC) to be a PCR template.

By further developing the methylation-dependent restriction enzyme-based method, in this paper, we present a novel, reliable (artifact-free), PCR-based (sequence-specific) approach to identify accurately the relative 6mA levels at any genomic site containing a GATC tetranucleotide sequence in any organism and organelle (Figure 1A’’). The protocol starts with the enzymatic digestion of a genomic DNA sample with DpnI; it is then followed by the ligation of a short adaptor DNA stretch, called linker, to the digested DNA fragments and terminates with a PCR amplification of the target site, using a forward primer that is simultaneously specific to a downstream part of the linker and a short mtDNA sequence adjacent to the target DpnI site. The PCR product can only be generated if the adenine at the target DpnI site was methylated (GA^Me^TC). Thus, unmethylated (GATC) DNA fragments are excluded from PCR amplification. For each sample, the total amount of mtDNA is measured with a restriction-independent internal control primer set annealing to another part of the mtDNA. With this internal control, we normalized the 6mA-sensitive, DpnI restriction-dependent PCR data to obtain relative 6mA levels in a given sample (Figure 1A,A’’).

To compare the detection sensitivity of the different restriction endonuclease-based 6mA detection methods, we prepared an in vitro assay, in which a fully 6mA methylated plasmid DNA was titrated by ten-fold dilutions with a completely unmethylated PCR-amplified DNA sample (Figure 1B,B’’). The DpnI digestion method performed the worst, as its detection sensitivity practically ceases if the average methylation of the tested DNA section is below 10% (Figure 1B). The MboI digestion method performed relatively well up to 1% methylation, but was unable to detect the DNA 6mA below that level (Figure 1B’). The method we introduced in this paper was the only one capable of detecting the DNA 6mA amounts below 0.01% (Figure 1B’’). Based on these results, it can be stated that the detection sensitivity of the different restriction endonuclease-based DNA 6mA detection methods decreases in the following order: DpnI digestion + Linker ligation > MboI digestion >> DpnI digestion. This is also important because, based on previous publications, the DNA 6mA level is below 1% per genome [11,13] and such low amounts can only be reliably detected with the DpnI digestion + linker ligation method described by us among the methods tested in this paper.

This PCR-based approach excludes any artifact that would result from RNA contamination or bacterial contamination since the primers are specific to the methylated target site selected. Without DpnI digestion or linker ligation, no PCR fragment was generated from genomic DNA templates (Figure 1C). In addition, digestion with methylation-sensitive MboI could also not result in quantitatively relevant PCR products (Figure 1D). The DpnI digestion and linker ligation method is readily applicable to any (nuclear and organellar) DNA fragment containing a GATC sequence in any organism and can even be translated to 5mC-dependent restriction enzymes, such as MspJI.

In evaluating the novel PCR-based approach to detect DNA adenine methylation, several inherent limitations emerge that necessitate careful consideration. Foremost among these is the technique’s strict dependence on the GATC tetranucleotide sequence, meaning that only specific genomic regions bearing this sequence can be assessed for 6mA methylation. This becomes particularly restrictive when DpnI cleavage sites are sparse or altogether missing in regions of interest, thus constricting the method’s broad applicability. Moreover, the technique is geared towards analyzing individual genetic loci, making it considerably labor-intensive when a panoramic survey of the genome’s methylation patterns is desired. Its sensitivity, although commendable in detecting methylation levels down to 0.01%, falters when confronted with ultra-low methylation levels below 0.001%, potentially overlooking or underestimating methylation in pivotal genomic stretches. Furthermore, the procedure’s intricate multi-step nature introduces avenues for variability and errors, making a consistent execution and rigorous optimization paramount. However, by integrating qPCR into the process, the sensitivity of the assay can be enhanced, offering a potential solution to some of these challenges.

### 3.2. N^6^-Adenine Methylation Increases Progressively with Age in the mtDNA of C. elegans

Using the method described above, we first explored that mtDNA isolated from the wild-type *C. elegans* strain is *N^6^*-adenine-methylated (Figure 2A,A’). Different mtDNA target sites (designated as *mitos 3* and *4*) were selected and analyzed at different stages of adulthood (at adult days 1, 4, 7, 9 and 12, since *C. elegans* lives for around 2 weeks), and all of these sites proved to be positive for DNA *N^6^*-adenine methylation. These results imply that essentially any part of the mitochondrial genome can be subjected to the DNA *N^6^*-adenine methylation process. However, the most exciting finding was the demonstration that relative 6mA levels at these sites gradually increase as the organism ages (Figure 2A,A’). The 6mA levels were relatively low at young (day 1) adulthood and elevated throughout the adult life span. Thus, the relative level of 6mA at a given mtDNA site is proportional to age; the older the animal, the higher the 6mA level at a given mtDNA site (Figure 2A,A’).

To confirm this observation, we performed an SMRTseq analysis on the mtDNA samples isolated at the adult stages of days 1 and 5 (Figure 3A,A’ and Appendix A). By identifying *N^6^*-methylated adenine sites and comparing the total 6mA levels of each chromosome with those of the mitochondrial genome within each age group (this excludes any artifact that may arise from variances in the amount of different DNA templates), *N^6^*-adenine methylation in the mtDNA relative to the nuclear genome was more robust in the aged sample. Similar results were obtained by a LC-MS/MS assay performed on isolated mtDNA samples (the method actually measures the total amounts of *N^6^*-methylated adenines in DNA samples completely digested to single nucleotides); the 6mA levels were significantly higher at the adult stage day 7 than day 1 (Figure 3B and Appendix A). We conclude that *N^6^*-adenine methylation in the *C. elegans* mtDNA increases with age.

We used the SMRTseq and LC-MS/MS methods because we wanted to confirm our results with independent methods. Nevertheless, we must draw attention to the fact that both of these methods have been strongly criticized recently [17,20], as they are often unable to specifically detect 6mA from sample DNA due to significant detection noise (nearby cytosine methylation) and bacterial DNA and RNA contamination. Based on these, we recommend that the results obtained with the SMRT and LC-MS/MS methods be treated with more caution.

### 3.3. mtDNA N^6^-Adenine Methylation Takes Place at a Lower Rate in Long-Lived Mutant Nematodes

We also assessed the rate at which mtDNA 6mA levels change during life span in long-lived *daf-2(−)* mutant animals (Figure 4D,D’). *daf-2* encodes a nematode insulin/IGF-1 (insulin-like growth factor) receptor, and worms defective for DAF-2 activity live two times longer than the wild type under conditions permitting reproductive growth [32,33]. We found that, in *daf-2(−)* mutants, *N^6^*-adenine methylation in the mitochondrial genome also gradually increases with age, and 6mA levels reach a similar maximal value at the end of life, relative to the wild type (Figure 4A,A’). However, the mtDNA *N^6^*-adenine methylation process occurred at a halved pace in *daf-2(−)* mutants as compared with the wild types (Figure 4B). We conclude that mtDNA 6mA levels serve as a signature of aging, thereby providing as a novel and reliable marker to determine age from DNA in this organism.

### 3.4. DAMT-1 DNA N^6^-Adenine Methyltransferase and NMAD-1 N^6^-Methyladenine Demethylase Influence mtDNA 6mA Levels

In the nuclear genome of *C. elegans*, methylation at the *N^6^*-position of adenine is suggested to be catalyzed by the enzyme DAMT-1 (DNA *N^6^*-adenine methyltransferase), whereas the demethylation of 6mA is mediated by the NMAD-1 enzyme (*N^6^*-methyladenine demethylase) [11]. DAMT-1 is known to respond to mitochondrial stress [34]. To examine whether NMAD-1 also participates in a mitochondrion-associated function, we generated a strain transgenic for a translational fusion NMAD-1::GFP reporter and stained transgenic animals with TMRE (tetramethylrhodamine ethyl ester), a red-orange fluorescent dye that is readily sequestered by active mitochondria (Figure 5A) [35]. The fluorescence microscopy analysis we conducted showed that NMAD-1 can be co-localized with TMRE, implying a partial existence of the enzyme in the mitochondria.

To further show the co-localization of NMAD-1::GFP and mitochondria with an independent method, we separated the cytoplasmic and mitochondrial cellular fractions from *C. elegans* by centrifugation, and then the two fractions were stained with a-GFP antibody for a Western blot analysis (Figure 5B). The experiment also confirmed that NMAD-1::GFP is specifically localized in the mitochondrial fraction compared to the cytoplasmic fraction.

Furthermore, the relative 6mA levels at a given mtDNA site were determined in *nmad-1(-)*- and *damt-1(-)*-mutant genetic backgrounds (Figure 5C). We found that both enzymes strongly influence 6mA levels in the mitochondrial genome. In the absence of NMAD-1 activity, the mtDNA 6mA levels were high already at the adult stage day 1. A similar 6mA level was detected at the mid-adult stages (at day 5). In contrast, DAMT-1 deficiency led to no upload with 6mA marks in the mtDNA at the early and mid-adult stages (at days 1 and 5). Thus, the enzymes that are involved in 6mA metabolism in the nuclear genome also drive the *N^6^*-adenine methylation/demethylation processes in the mitochondrial genome.

### 3.5. The Presence of 6mA Is also Detectable in the mtDNA of Drosophila and Dog, and Its Levels Increase with Age

To assess whether *N^6^*-adenine methylation in the mitochondrial genome is specific to nematodes or represents an evolutionarily conserved process, we subsequently applied the method shown in Figure 1A’’ to DNA samples isolated from the fruit fly *Drosophila melanogaster*. Primer pairs specific to different sites of *Drosophila* mtDNA were used in the PCR amplification step to avoid any artifact. Significant levels of mtDNA 6mA were detected at different sites (called *mitos 2* and *4*) in this genetic model system too (Figure 6A,A’). Similar to *C. elegans*, 6mA levels in the fly mitochondrial genome also increased during the adult life span (Figure 6A,A’). A significant drop in 6mA levels was evident only at the latest adult days, which may result from the fact that mitochondrial activity and function are strongly impaired in aged cells [36,37].

Subsequently, we examined the *Drosophila* Tet DNA *N^6^*-adenine methyltransferase and Mt2 *N^6^*-methyladenine demethylase that are orthologous to nematode NMAD-1 and DAMT-1, respectively [12]. Consistent to what we found in *C. elegans* (Figure 5B), the downregulation of *Tet* significantly increased, while the silencing of *Mt2* was considerably decreased, in relation to 6mA levels in mtDNA at a given adult stage as compared with the control (*w^1118^*) (Figure 6B and Appendix A).

Then, we examined the existence of 6mA in mtDNA isolated from dogs. Contrary to other eukaryotic taxa, in mammals, *N^6^*-adenine methylation is proposed to have a gene-silencing function [13]. However, we also uncovered a gradual increase in 6mA levels in the dog mitochondrial genome over the adult life span (Figure 7A,A’). These results are supported by a qPCR analysis showing a higher amount of 6mA marks in aged animals than those found in young individuals (Figure 7B,C).

## 4. Discussion

In this study, we presented a reliable (sequence-specific, PCR-based) method by which one can unambiguously identify 6mA sites in the mitochondrial (and potentially nuclear) genomes of diverse animal taxa and determine the relative 6mA levels at a given site containing a GATC tetranucleotide sequence (Figure 1A’’). The method relies on the PCR-directed amplification of the target sequence digested by the 6mA-dependent DpnI restriction endonuclease and ligated to a linker stretch (the forward primer is simultaneously specific to the linker and adjacent genomic sequence). The results obtained in this paper have at least three important implications. First, *N^6^*-adenine methylation has been recently discovered in the nuclear genomes of various animal taxa and in mammalian mtDNA [11,12,13,16]. This was particularly important in species largely devoid of cytosine methylation, such as *C. elegans* and *D. melanogaster*, as in these organisms 6mA is now known to represent the sole DNA methylation-based epigenetic mark. Recent conflicting data, however, seriously argue against the presence of *N^6^*-adenine methylation in animal and organellar genomes [17,18,19]. Coupled with the ligation of a linker fragment to the digested site, the PCR-based method we presented in this paper (Figure 1A’’) is capable of identifying 6mA marks in a sequence-specific manner, thereby generating no artifact due to significant detection noise (nearby cytosine methylation) and bacterial DNA and RNA contamination. Using this technique, we showed the presence of *N^6^*-adenine methylation in the mtDNA of three divergent animal species, the nematode *Ceanorhabditis elegans*, insect *Drosophila melanogaster* and dog *Canis lupus* (Figure 2, Figure 3, Figure 4, Figure 5, Figure 6 and Figure 7). Thus, we showed that adenine can also be methylated on the *N^6^* position in the mitochondrial genome to form a conserved epigenetic mark. This observation may also be important because the existence of mitochondrial non-CpG methylation has recently been questioned [38]; so, it is possible that the mitochondrial 6mA modification is the main mitochondrial epigenetic modification at least in the studied organisms.

Second, we demonstrated that the mtDNA becomes progressively *N^6^*-methylated on adenines in *C. elegans*, *Drosophila* and dogs over the adulthood (Figure 1, Figure 2, Figure 3, Figure 4, Figure 5, Figure 6 and Figure 7). In nematodes, the rate at which *N^6^*-adenine methylation occurs in long-lived *daf-2(−)*-mutant animals is around two times slower than that in the wild type (Figure 4B). This correlates nicely with the double longevity of *daf-2(−)* mutants. Thus, 6mA accumulation in the mitochondrial genome is proportional with the actual life stage of the animal; the higher the level of this epigenetic mark at a given site, the older the organism. Interestingly, both wild-type and *daf-2(−)*-mutant *C. elegans* strains displayed a similar maximum level of 6mA marks at a given site of mtDNA at the end of adulthood, but the latter required proportionately more time to reach this level (Figure 4B). Therefore, the dynamics of 6mA accumulation in the mtDNA is slower in these long-lived mutant animals. Based on these data, we propose that *N^6^*-adenine methylation may serve as a mitochondrion-specific epigenetic clock (Figure 8). The gradual increase in mtDNA 6mA levels during aging is so robust and evolutionarily conserved that it can serve as a reliable epigenetic mark to determine age from a given tissue sample. Nevertheless, further studies should reveal why 6mA gradually accumulates with age in the mtDNA.

Third, *N6*-adenine methylation and demethylation in the mtDNA appears to be governed, at least in part, by the same enzymatic machinery that catalyzes the processes in the nuclear genome. In *C. elegans*, DAMT-1 has been identified as a putative DNA *N^6^*-adenine methyltransferase converting adenine into 6mA [11]. In mutant nematodes deficient in DAMT-1 activity, *N^6^*-adenine methylation failed to occur in the mitochondrial genome at different adult stages (Figure 5C). A similar effect was observed when the *Drosophila* DNA *N^6^*-adenine methyltransferase-encoding *Mtt2* gene [12] was downregulated (Figure 6B). Moreover, the *C. elegans N^6^*-methyladenine demethylase NMAD-1 [11] was readily co-localized with a mitochondrial marker (Figure 5A,B), and its deficiency was associated with increased mtDNA 6mA levels relative to the control (Figure 5C). In *Drosophila*, the downregulated *Tet* gene coding for *N^6^*-methyladenine demethylase [12] also led to increased 6mA levels in the mtDNA compared with the wild type (Figure 6B). Further studies should prove the mitochondrial translocation of these enzymes involved in DNA *N^6^*-adenine methylation or *N^6^*-methyladenine demethylation.

Recently, a number of age-determining biomarkers and methods have been published that can be used as an aging clock, such as Horvath’s clock and other cytosine methylation-based aging clocks [8], transcriptome-based aging clocks (such as BiT age [39]), inflammatory aging clock (such as iAge) [40], and the metabolomic aging clock [41]. The greatest disadvantage of these methods is that they are all based on very expensive and time-consuming techniques and this makes them difficult to use widely. One of the major advantages of the mtDNA 6mA-detecting PCR-based method we presented in this paper is that it is inexpensive and fast to perform, providing an opportunity for the method to spread beyond scientific research into clinical diagnostics and forensics. In general, the method may serve as the basis of epigenetic DNA diagnostics to identify routinely the methylation status of disease genes.

## Figures and Tables

**Figure 1 ijms-24-14858-f001:**
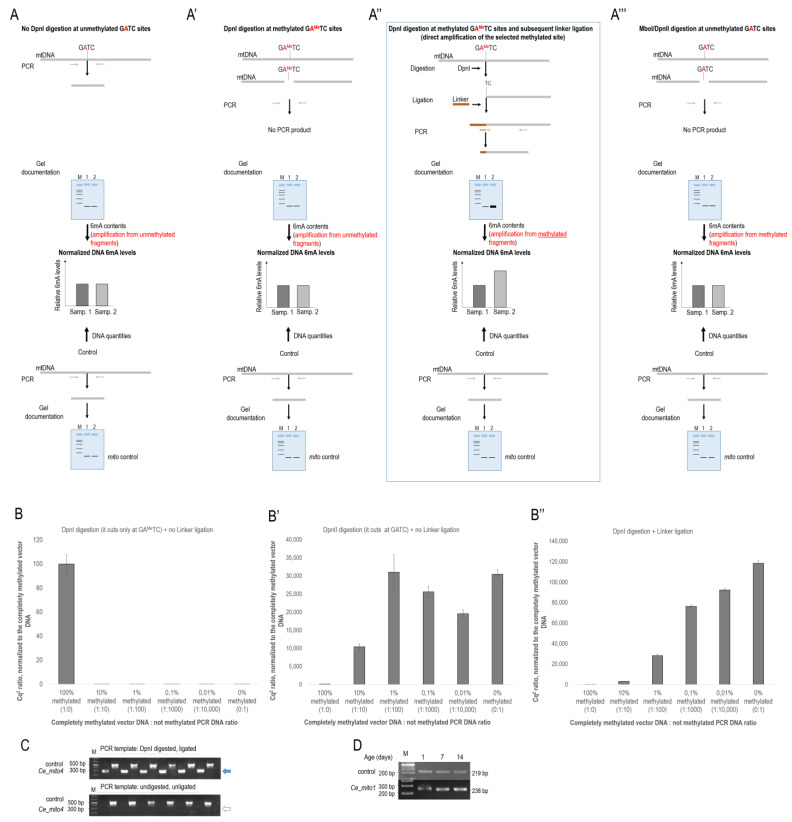
**Testing 6mA levels by different PCR-based methods.** (**A**) Scheme of the DpnI treatment at (unmethylated) GATC sequences. The enzyme cannot digest this sequence; so, the target site remaining intact becomes PCR-amplified. The amount of PCR product is proportional to the amount of template DNA applied. (**A’**) DpnI digestion at (methylated) the GA^Me^TC site. The digested sequence cannot serve as a template for PCR amplification. At this site, PCR reaction amplifies only the intact (unmethylated) GATC sequences found in the tissue sample. Because *N^6^*-adenine methylation occurs at a very low rate for a given genomic (mtDNA) site, this method is incapable of identifying accurately 6mA levels in this sequence. (**A’’**) Scheme of the method we developed to determine accurately the relative 6mA levels at specific genomic sites containing a GATC tetranucleotide sequence (framed). The genomic (mtDNA) DNA is digested with DpnI restriction endonuclease (this methylation-dependent enzyme cuts a tetranucleotide GA^Me^TC sequence when A is methylated). An adaptor DNA stretch, called linker, is then ligated to the digested genomic DNA fragments. The PCR amplification of the selected mtDNA site is mediated by a forward primer that is specific to both linker (brown part of the left primer) and a short mtDNA sequence adjacent to the selected GA^Me^TC site (grey part of the left primer). This primer makes possible a direct amplification of the methylated (digested) GATC sequence. (**A’’’**) MboI/DpnII digestion of a selected GATC sequence. MboI and DpnII cut the target sequence when A is unmethylated. Hence, a subsequent PCR reaction can use the methylated, i.e., undigested, sequences as a template for amplification. However, enzymatic reactions are generally incomplete, thereby always leaving a few unmethylated sequences intact. These sequences compete with the methylated ones as a template for amplification. Because the level of *N^6^*-methylation at a given site is usually very low, this competition makes 6mA level identification rather unreliable. (**A**–**A’’’**) After PCR amplification, the products are evaluated by standard gel electrophoresis. M indicates the molecular weight marker, and 1 and 2 represent different samples. Bottom panels: internal control, which is used to quantify the total amount of DNA that was included in the test. In this case, the PCR reaction is independent of the restriction endonuclease; hence, the amount of PCR products is proportional to the amount of DNA samples. Middle panels: 6mA levels are normalized to the corresponding control samples to obtain the relative 6mA contents. (**B**–**B’’**) To validate this novel method shown in panel (**A’’**), we assessed the levels of methylated and unmethylated adenines in an in vitro test. We isolated plasmid DNA from a DH5α *E. coli* bacterial strain and measured the 6mA levels at a given site. This was determined as a full (100%) methylation level. Then, the same plasmid site was PCR-amplified and determined as a zero (0%) methylation level (PCR amplification is devoid of methylation). We subsequently generated a concentration gradient by adding the PCR products to the plasmid solution in gradually increasing quantities and measured the DNA amounts by using qPCR (the Cq^2^ values were determined). The results shown below clearly demonstrate that our method presented in this paper (**A’’**), but not the simple DpnI (**A**) or DpnII/MboI digestion (**A’’’**) without linker ligation and subsequent qPCR amplifications, is sensitive enough to measure 6mA levels accurately at the selected site. The DpnI digestion of genomic DNA, linker ligation to genomic fragments and a subsequent PCR amplification of the target site with a specific forward primer (as shown in panel (**A’’**)) result in accurate 6mA levels: the higher the 6mA level in the sample at a given site, the higher the amount of the ligated fragments and, as a consequence, the higher the amount of the PCR product (the Cq^2^ value). (**C**) Control experiments with undigested and unligated genomic samples. Upper panel: genomic DNA was digested with DpnI, and then the digested DNA stretches were ligated to linker. Control and *mito4*-specific PCR fragments are visible. The latter is indicated by a blue arrow. Lower panel: genomic DNA was not digested and ligated. Only the control PCR fragments are visible, whereas *mito4*-specific PCR fragments are not (the place where they should migrate into the gel is shown by an empty arrow). (**D**) MboI is a methylation-sensitive restriction enzyme that cuts the GATC sequence when A is unmethylated. Only the undigested, intact (methylated GA^Me^TC) fragments serve as a template for a subsequent PCR amplification. The results show no differences among the samples isolated at different adult stages.

**Figure 2 ijms-24-14858-f002:**
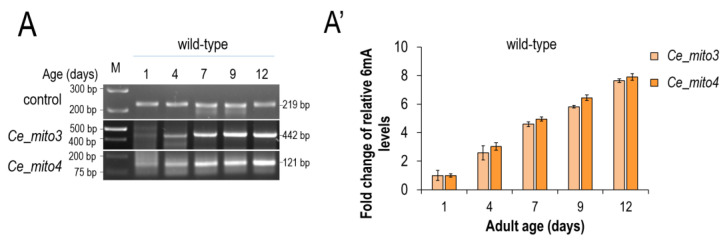
***N^6^*-methyladenine levels in the *C. elegans* mitochondrial genome gradually increase with age.** (**A**) The 6mA progressively accumulates at different mtDNA sites (*mito 3* and *mito 4*) during aging. (**A’**) Quantification of the relative mtDNA 6mA levels at different adult stages. Bars indicate ±S.D.; each comparison reveals ***: *p* < 0.001 significance. For statistics, see Appendix A.

**Figure 3 ijms-24-14858-f003:**
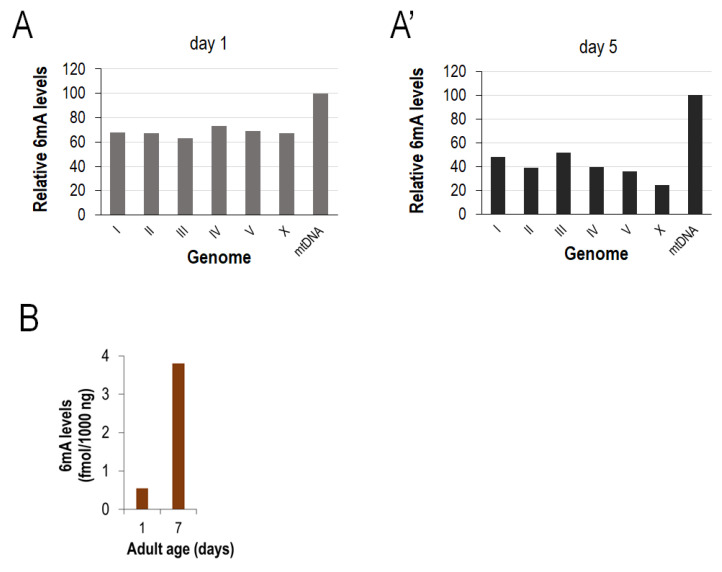
**SMRTseq and LC-MS/MS analyses confirm that *N^6^*-methyadenine levels in the *C. elegans* mtDNA increase with age.** (**A**,**A’**) Determined by the SMRT sequencing of genomic DNA samples isolated at different adult ages, the global 6mA levels of each chromosome and mtDNA at adult days 1 (**A**) and 5 (**A’**). The absolute mtDNA levels were considered as 100. Relative to this level, the chromosomal 6mA levels are lower in the old sample (day 5) compared with the young one (day 1). (**B**) LC-MS/MS analysis showing a significantly higher amount of 6mA marks in the mtDNA isolated from aged (7-day-old) adults compared with young (1-day-old) ones. For data, see Appendix A.

**Figure 4 ijms-24-14858-f004:**
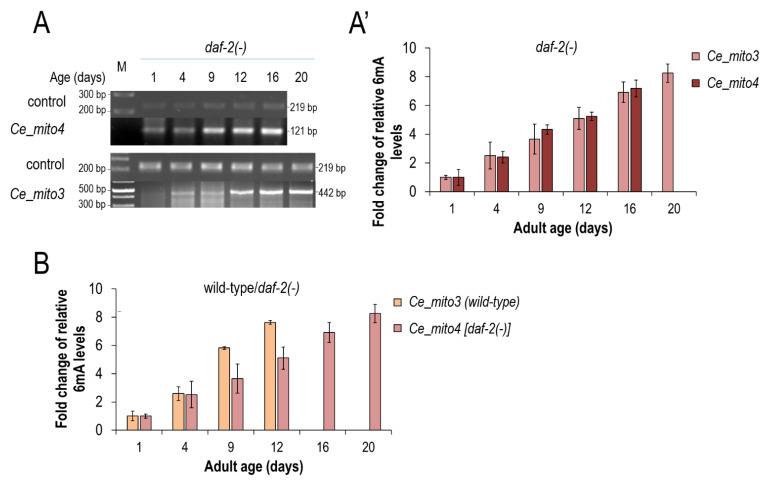
**In long-lived *daf-2(-)*-mutant nematodes, *N^6^*-adenine methylation in the mtDNA increases with a slower rate as compared to the wild type.** (**A**) In *daf-2(-)*-mutant nematodes, which live two times longer than the wild type, the *N^6^*-adenine methylation process at two selected mtDNA sites also gradually increases with age. (**A’**) Quantification of mtDNA 6mA levels in a *daf-2(-)*-mutant background at different adult stages. (**B**) Comparison of the age-dependent dynamics of *N^6^*-adenine methylation between the wild-type and *daf-2(-)*-mutant animals. The age-normalized rates at which the 6mA levels increase in these two genetic backgrounds are highly similar to each other. At the latest adult stages, the mtDNA 6mA levels are similar between the two strains. Bars represent ±S.D.; For statistics, see Appendix A.

**Figure 5 ijms-24-14858-f005:**
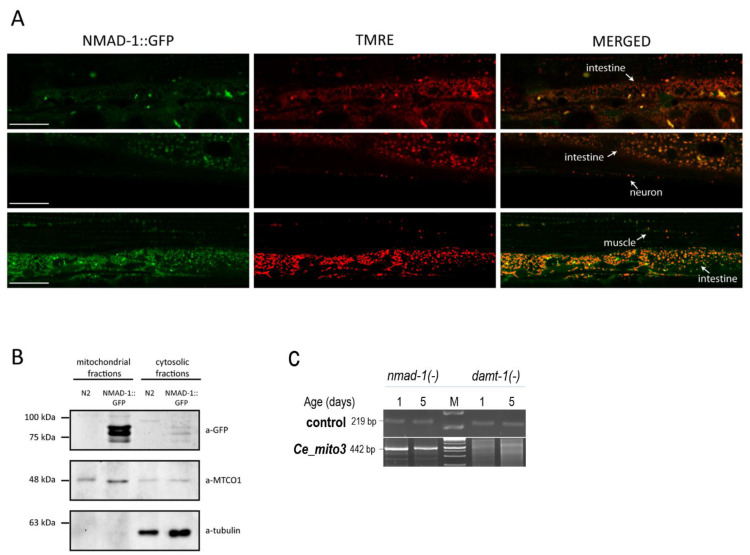
**DAMT-1, which is a putative DNA *N^6^*-adenine methyltransferase, and NMAD-1 *N^6^*-methyladenine demethylase influence 6mA levels in the *C. elegans* mtDNA.** (**A**) Confocal images of animals transgenic for a NMAD-1::GFP translational reporter and stained with the mitochondrial dye TMRE. NMAD-1::GFP (green) co-localizes with TMRE (red) in different tissues (indicated with arrows). Scale bars: 20 µm. (**B**) Western blot analysis for the detection of NMAD-1::GFP using an antibody against GFP (a-GFP). MTCO1 and tubulin were used as loading controls for the mitochondrial and cytoplasmic fractions, respectively. Fractions from wild-type (N2) animals were used as the controls to verify that a-GFP specifically detects NMAD-1::GFP. (**C**) Both NMAD-1 and DAMT-1 strongly influence 6mA levels in mtDNA. NMAD-1 deficiency leads to high 6mA levels in the mtDNA even at an early (day 1) adult stage. Defects in DAMT-1 activity highly reduce 6mA levels in the mtDNA at the early (day 1) and mid- (day 5) adult stages.

**Figure 6 ijms-24-14858-f006:**
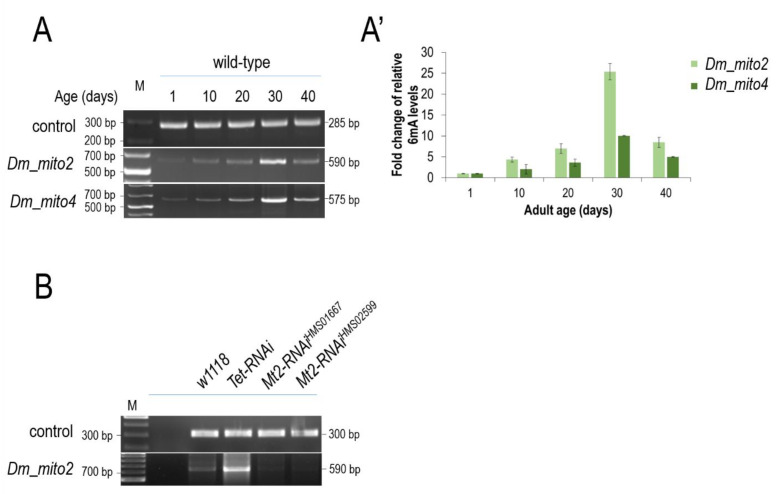
**mtDNA *N^6^*-adenine methylation in *Drosophila* also gradually increases with age.** (**A**) The 6mA at a given mtDNA site progressively accumulates during *Drosophila* aging. (**A’**) Quantification of 6mA levels in the *Drosophila* mtDNA at different adult stages. (**B**) Downregulated *Drosophila* Tet *N^6^*-methyladenine demethylase increases 6mA levels in the mtDNA at early adult stages as compared with the control. The decreased activity of fly Mt2 *N^6^*-adenine methyltransferase enormously lowers 6mA levels in the mtDNA relative to the control. Two independent Mt2-RNAi clones were examined. DNA was prepared from the head samples of males at the adult stage of day 10. Bars represent ±S.D.; For statistics, see Appendix A.

**Figure 7 ijms-24-14858-f007:**
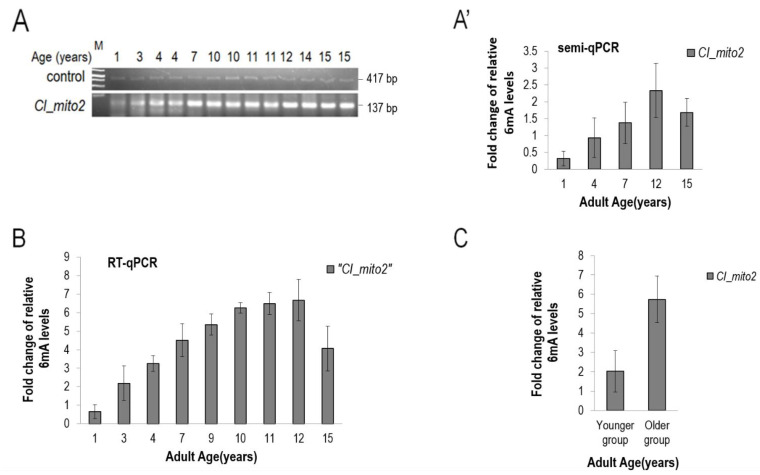
***N^6^*-adenine methylation in the mtDNA increases gradually with age in dogs.** (**A**) Relative 6mA levels in the mtDNA of dogs at different adult ages. The 6mA levels gradually increase during the adult life span. (**A’**) Quantification of semi-qPCR data. Samples were analyzed in triplicate. (**B**) Quantification of 6mA levels in the dog mtDNA at different adult ages by qPCR. The same samples as those shown in panel (**A’**) were analyzed. (**C**) Comparing 6mA levels in the dog mtDNA between young and aged groups. Bars represent ±S.D.; For statistics, see Appendix A.

**Figure 8 ijms-24-14858-f008:**
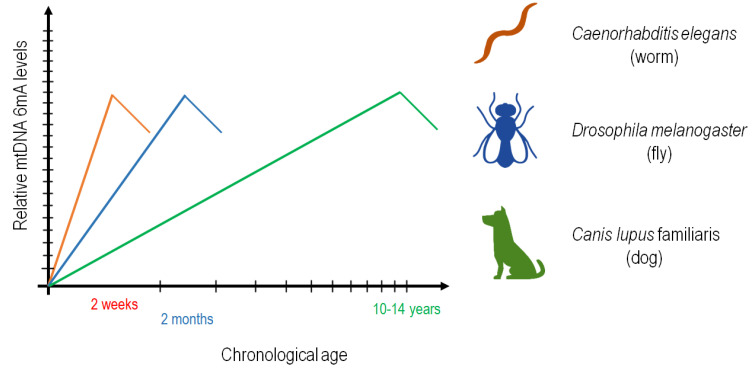
**Model showing the correlation between age and relative 6mA levels in mtDNA.** According to the model, mtDNA 6mA levels gradually increase during the adult life span. At advanced ages, mtDNA 6mA levels begin to decrease, which may result from the fact that mitochondrial activity and function are strongly impaired in aged cells.

## Data Availability

All data supporting the findings of this study are provided within the paper and it’s Appendix A. Any data are available from the authors upon request. Raw SMRT-sequencing reads of the 1 day old and 5 days old animals are available in the NCBI Sequence Read Archive (SRA) database with the BioProject accession code PRJNA682481. All data, code, and materials used in the analysis are available upon reasonable request for collaborative studies regulated by materials/data transfer agreements (MTA/DTAs) to the corresponding author (vellai.tibor@ttk.elte.hu).

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
