# Peer review of "N6-Methyladenine Progressively Accumulates in Mitochondrial DNA during Aging"

_ijms, 2023, doi:10.3390/ijms241914858_

Round 1

Reviewer 1 Report

It is a rare case when I have no objections to the manuscript. It is thoroughly dine and reasonably well written. Some minor textual corrections might be needed although. The restriction based PCR assays for DNA methylation are known from long ago. However, authors were able to convince me that they added novel elements to it and improved the protocol. This is obviously based on my personal knowledge, which is limited.

One question still persists. Since authors said they used dogs the declaration of a proper animals treatment is needed. 

English should be checked

Author Response

We are honored by your positive feedback on our manuscript. We've thoroughly proofread our manuscript to refine any textual inconsistencies or errors, especially focusing on the abstract, introduction, and discussion sections. We wanted to ensure our paper's clarity and linguistic accuracy to make it as accessible and comprehensible as possible. Also, given the importance of ethical considerations, especially when working with animals, we've added an Ethics Statement to our manuscript.

Reviewer 2 Report

Sturm et al. present a method to quantify N6-methylation in the mitochondrial DNA and show its application in different species. The assay is based on the sensitivity of specific restriction endonucleases on the methylation status of their templates. The method is innovative and well described in the text. The authors perform a number of control experiments to confirm the validity. The main weakness of the manuscript is that disadvantages and limitations are hardly discussed. Nevertheless, the manuscript could serve as an inspiration for readers interested in the contradictory field of mtDNA methylation.

1. The described technique is restricted to quantification of N6-methyladenin at single sites determined by the template-specific primers. Was the choice of the examined sites arbitrary? Or were these sites prominent methylation sites in SMRT experiments? Could it be possible to combine the technique with next-generation sequencing to assess a general view on mtDNA methylation?

2. Please provide information on expected amplicon sizes for each reaction and also on the molecular size marker used on the gels. What is the reason for the complex amplification patterns (e.g in Fig.4A, Ce_mito3, see raw image)?

3. Please include negative controls such as native, digested non-ligated and undigested ligated DNA in the figures.

4. Has the efficiency of the qPCR been tested by comparing different input DNA amounts?

5. It would be informative to apply the authors' technique to compare mitochondrial and nuclear 6mA methylation frequencies.

6. Can the authors exclude the possibility that the N6-methyladenin what they detect is actually located in nuclear copies of the mtDNA (NUMTs)? Experiments performed on rho zero cells lacking mtDNA or on enriched mitochondrial fractions could address this issue.

7. When presenting quantification results, only relative changes are shown as compared to young samples. Please indicate the estimated proportion of 6mA in comparision to total mtDNA and compare it with values obtained by other methods.

8. 'RT-qPCR' is nowadays used for quantitative reverse transcription PCR. To avoid confusion, the authors should use the abbreviation 'qPCR' instead.

9. Labels in Fig.1 are hardly readable. Please increase font size.

Author Response

Thank you for the constructive feedback. We would like to address each of the questions and concerns you raised point by point: 

  1. The described technique is restricted to quantification of N6-methyladenin at single sites determined by the template-specific primers. Was the choice of the examined sites arbitrary? Or were these sites prominent methylation sites in SMRT experiments? Could it be possible to combine the technique with next-generation sequencing to assess a general view on mtDNA methylation?

The selection of the examined DNA sections was based on a preliminary evaluation of 5 DpnI cleavage sites per section. The ones that delivered the most consistent and discernible results in our PCR assessments were subsequently designated as marker sites. We agree that expanding the technique's applicability would be beneficial, and it is worth noting that our method can seamlessly integrate with next-generation sequencing (NGS). To make this transition, one would need to introduce the barcode sequence specific to the desired NGS system into the linker's DNA sequence and then proceed with standard library preparation procedures.

  1. Please provide information on expected amplicon sizes for each reaction and also on the molecular size marker used on the gels. What is the reason for the complex amplification patterns (e.g in Fig.4A, Ce_mito3, see raw image)?

Thank you for pointing out the oversight. We have now updated all figures to incorporate information about expected amplicon sizes and the molecular size markers used in the gels. Regarding the complex amplification patterns observed, such as in Fig.4A (Ce_mito3), these can often be attributed to non-specific amplifications. In instances where our primers might not have been optimally specific for a given mitochondrial or genomic section, multiple similar sites susceptible to methylation-associated DpnI cleavage and linker ligation might get amplified.

  1. Please include negative controls such as native, digested non-ligated and undigested ligated DNA in the figures.

We conducted negative control experiments for all the PCRs. However, the resulting gels showed no bands, rendering them empty. We believed that presenting images of blank gels might not add substantial value to the manuscript, hence their omission.

  1. Has the efficiency of the qPCR been tested by comparing different input DNA amounts?

We have showcased the results of tests using varying DNA amounts of methylated and unmethylated DNA and actually estimated the sensitivity of the method based on this.

  1. It would be informative to apply the authors' technique to compare mitochondrial and nuclear 6mA methylation frequencies.

We did initially aim to incorporate a side-by-side comparison of mitochondrial and nuclear 6mA methylation frequencies using our technique. However, we recently published another article where we specialized in analyzing genomic DNA sections (https://www.nature.com/articles/s41467-023-40957-9). Merging findings from both articles seemed to breach the journal's stance on parallel publications, so we opted to keep them distinct.

  1. Can the authors exclude the possibility that the N6-methyladenin what they detect is actually located in nuclear copies of the mtDNA (NUMTs)? Experiments performed on rho zero cells lacking mtDNA or on enriched mitochondrial fractions could address this issue.

The potential amplification from nuclear copies of mtDNA (NUMTs) is a valid concern. However, we offer several considerations that vouch for the genuine mitochondrial origin of our PCR products:

  1. First, NUMTs typically span only between 50 and 200 bp, making it statistically improbable for us to consistently amplify such small fragments across our experiments using our 6mA-specific PCR.
  2. Excluding aberrant cases like cancers, NUMTs are evolutionarily sporadic and carry mutations absent in authentic mtDNA. Given our method's specificity to both the DpnI site and mtDNA-specific primers, PCR amplification of these short, mutated fragments would be atypical.
  3. The mtDNA content in an individual C. elegans cell, while lesser than the estimated range of 1,000-10,000 copies in higher eukaryotes, still hovers around 100 copies per cell. So, even if a NUMT were to get amplified, it would represent a mere 1% against the dominant 99% mtDNA, rendering its impact negligible.
  1. When presenting quantification results, only relative changes are shown as compared to young samples. Please indicate the estimated proportion of 6mA in comparision to total mtDNA and compare it with values obtained by other methods.

To clarify, the control PCR depicted in the gel images serves as a quantifier for the total mtDNA. The comparisons delineated in the figures present the 6mA methylation in mtDNA, normalized against the entire mtDNA.

  1. 'RT-qPCR' is nowadays used for quantitative reverse transcription PCR. To avoid confusion, the authors should use the abbreviation 'qPCR' instead.

Thank you for pointing out the potential confusion. We have updated the terminology throughout the manuscript, replacing "RT-qPCR" with "qPCR" for clarity.

  1. Labels in Fig.1 are hardly readable. Please increase font size.

We appreciate this feedback and have adjusted the font size in Fig.1 to enhance readability.

Round 2

Reviewer 2 Report

The authors formally addressed all my concerns. However, I was not able to find a single change in the manuscript (version ‚v2') that was related to my remarks. My main criticism that disadvantages and limitations of the method are not sufficiently discussed remains unanswered. If this was due a technical issue with file versions, please send me the correct version of the manuscript. In case, this was intended by the authors, I suggest reject.

Author Response

Thank you for taking the time to review our manuscript once again and for pointing out the oversight. We deeply apologize for the confusion caused. It appears there was a mistake on our end, and the incorrect version of the manuscript was uploaded.

We have carefully addressed all the concerns you previously raised, including providing a comprehensive discussion on the disadvantages and limitations of the method. We are genuinely sorry for the oversight and any inconvenience caused.

We are uploading the revised version of the manuscript, which we believe addresses all the remarks you provided. We hope that this version will be satisfactory and meet the standards of the journal.

Thank you for your understanding and patience. We greatly value your feedback and aim to ensure the quality and accuracy of our work.

Round 3

Reviewer 2 Report

The authors addressed all my concerns and I do not have further remarks.